# Promoting Saliency From Depth: Deep Unsupervised RGB-D Saliency Detection

**Wei Ji**[1], **Jingjing Li**[1,*], **Qi Bi**[2], **Chuan Guo**[1], **Jie Liu**[3], **Li Cheng**[1]
[1]University of Alberta, Canada  [2]Wuhan University, China  [3]Dalian University of Technology, China
`{wji3, jingjin1, cguo2, lcheng5}@ualberta.ca`

## Abstract

Growing interests in RGB-D salient object detection (RGB-D SOD) have been witnessed in recent years, owing partly to the popularity of depth sensors and the rapid progress of deep learning techniques. Unfortunately, existing RGB-D SOD methods typically demand large quantity of training images being thoroughly annotated at pixel-level. The laborious and time-consuming manual annotation has become a real bottleneck in various practical scenarios. On the other hand, current unsupervised RGB-D SOD methods still heavily rely on handcrafted feature representations. This inspires us to propose in this paper a *deep unsupervised RGB-D saliency detection* approach, which requires *no* manual pixel-level annotation during training. It is realized by two key ingredients in our training pipeline. First, a depth-disentangled saliency update (DSU) framework is designed to automatically produce pseudo-labels with iterative follow-up refinements, which provides more trustworthy supervision signals for training the saliency network. Second, an attentive training strategy is introduced to tackle the issue of noisy pseudo-labels, by properly re-weighting to highlight the more reliable pseudo-labels. Extensive experiments demonstrate the superior efficiency and effectiveness of our approach in tackling the challenging unsupervised RGB-D SOD scenarios. Moreover, our approach can also be adapted to work in fully-supervised situation. Empirical studies show the incorporation of our approach gives rise to notably performance improvement in existing supervised RGB-D SOD models.

## 1 Introduction

The recent development of RGB-D salient object detection, *i.e.*, RGB-D SOD, is especially fueled by the increasingly accessible 3D imaging sensors (Giancola et al., 2018) and their diverse applications, including image caption generation (Xu et al., 2015), image retrieval (Ko et al., 2004; Shao & Brady, 2006) and video analysis (Liu et al., 2008; Wang et al., 2018), to name a few. Provided with multi-modality input of an RGB image and its depth map, the task of RGB-D SOD is to effectively identify and segment the most distinctive objects in a scene.

The state-of-the-art RGB-D SOD approaches (Li et al., 2021a; Ji et al., 2020b; Chen et al., 2020b; Li et al., 2020b; 2021b) typically entail an image-to-mask mapping pipeline that is based on the powerful deep learning paradigms of *e.g.*, VGG16 (Simonyan & Zisserman, 2015) or ResNet50 (He et al., 2016). This strategy has led to excellent performance. On the other hand, these RGB-D SOD methods are fully supervised, thus demand a significant amount of pixel-level training annotations. This however becomes much less appealing in practical scenarios, owing to the laborious and time-consuming process in obtaining manual annotations. It is therefore natural and desirable to contemplating unsupervised alternatives. Unfortunately, existing unsupervised RGB-D SOD methods, such as global priors (Ren et al., 2015), center prior (Zhu et al., 2017b), and depth contrast prior (Ju et al., 2014), rely primarily on handcrafted feature representations. This is in stark contrast to the deep representations learned by their supervised SOD counterparts, which in effect imposes severe limitations on the feature representation power that may otherwise benefit greatly from the potentially abundant unlabeled RGB-D images.

These observations motivate us to explore a new problem of *deep unsupervised RGB-D saliency detection*: given an unlabeled set of RGB-D images, deep neural network is trained to predict saliency without any laborious human annotations in the training stage. A relatively straightforward idea is to exploit the outputs from traditional RGB-D method as pseudo-labels, which are internally employed

---

*Corresponding author

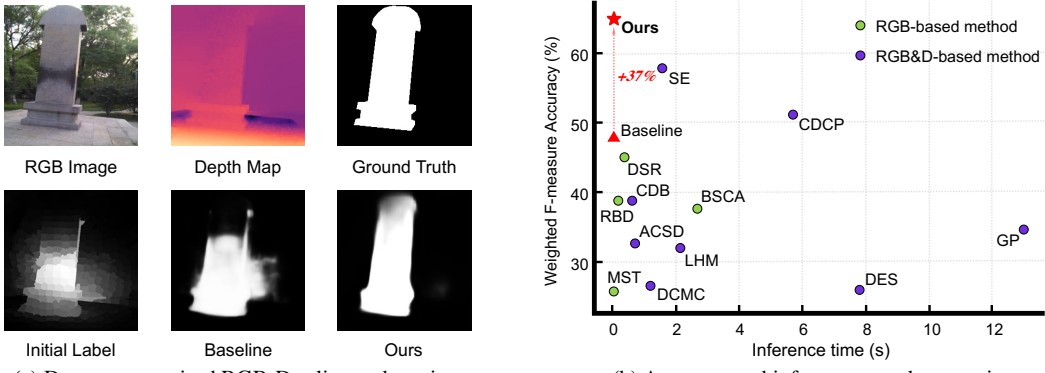

(a) Deep unsupervised RGB-D saliency detection

(b) Accuracy and inference speed comparison

Figure 1: (a) An illustration of deep unsupervised RGB-D saliency detection. 'Initial label' is generated by a traditional method. 'Baseline' shows the saliency map generated by saliency network trained with initial pseudo-labels. 'Ours' shows our final results. (b) Efficiency and effectiveness comparison over a wide range of unsupervised SOD methods on the NLPR benchmark. **Promoting Saliency From Depth:** our approach achieves a large-margin improvement over the baseline, by engaging depth information to improve pseudo-labels in the training process, without introducing additional computational cost during inference, shown in red arrow.

to train the saliency prediction network ('baseline'). Moreover, the input depth map may serve as a complementary source of information in refining the pseudo-labels, as it contains cues of spatial scene layout that may help in exposing the salient objects. Nevertheless, practical examination reveals two main issues: *(1) Inconsistency and large variations in raw depth maps*: as illustrated in Fig. 1 (a), similar depth values are often shared by a salient object and its surrounding, making it very difficult in extracting the salient regions from depth without explicit pixel-level supervision; *(2) Noises from unreliable pseudo-labels*: unreliable pseudo-labels may inevitably bring false positive into training, resulting in severe damage in its prediction performance.

To address the above challenges, the following *two* key components are considered in our approach. First, a depth-disentangled saliency update (DSU) framework is proposed to iteratively refine & update the pseudo-labels by engaging the depth knowledge. Here a depth-disentangled network is devised to explicitly learn the discriminative saliency cues and non-salient background from raw depth map, denoted as saliency-guided depth $D_{Sal}$ and non-saliency-guided depth $D_{NonSal}$, respectively. This is followed by a depth-disentangled label update (DLU) module that takes advantage of $D_{Sal}$ to emphasize saliency response from pseudo-label; it also utilizes $D_{NonSal}$ to eliminate the background influence, thus facilitating more trustworthy supervision signals in training the saliency network. Note the DSU module is not engaged at test time. Therefore, at test time, our trained model takes as input only an RGB image, instead of involving both RGB and depth as input and in the follow-up computation. Second, an attentive training strategy is introduced to alleviate the issue of noisy pseudo-labels; it is achieved by re-weighting the training samples in each training batch to focus on those more reliable pseudo-labels. As demonstrated in Fig. 1 (b), our approach works effectively and efficiently in practice. It significantly outperforms existing unsupervised SOD methods on the widely-used NLPR benchmark. Specifically, it improves over the baseline by 37%, a significant amount without incurring extra computation cost. Besides, the test time execution of our approach is at 35 frame-per-second (FPS), the fastest among all RGB-D unsupervised methods, and on par with the most efficient RGB-based methods. In summary, our main contributions are as follows:

- To our knowledge, our work is the first in exploring deep representation to tackle the problem of unsupervised RGB-D saliency detection. This is enabled by two key components in the training process, namely the DSU strategy to produce & refine pseudo-labels, and the attentive training strategy to alleviate the influence of noisy pseudo-labels. It results in a light-weight architecture that engages only RGB data at test time (*i.e.,* w/o depth map), achieving a significant improvement without extra computation cost.

- Empirically, our approach outperforms state-of-the-art unsupervised methods on four public benchmarks. Moreover, it runs in real time at 35 FPS, much faster than existing unsupervised RGB-D SOD methods, and at least on par with the fastest RGB counterparts.

- Our approach could be adapted to work with fully-supervised scenario. As demonstrated in Sec. 4.4, augmented with our proposed DSU module, the empirical results of existing RGB-D SOD models have been notably improved.

## 2 RELATED WORK

Remarkable progresses have been made recently in salient object detection (Ma et al., 2021; Xu et al., 2021; Pang et al., 2020b; Zhao et al., 2020b; Tsai et al., 2018; Liu et al., 2018), where the performance tends to deteriorate in the presence of cluttered or low-contrast backgrounds. Moreover, promising results are shown by a variety of recent efforts (Chen et al., 2021a; Luo et al., 2020; Chen et al., 2020a; Zhao et al., 2022; Desingh et al., 2013; Li et al., 2020a; Liao et al., 2020; Shigematsu et al., 2017) that integrate effective depth cues to tackle these issues. Those methods, however, typically demand extensive annotations, which are labor-intensive and time-consuming. This naturally leads to the consideration of unsupervised SODs that do not rely on such annotations.

Prior to the deep learning era (Wang et al., 2021; Zhao et al., 2020a), traditional RGB-D saliency methods are mainly based on the manually-crafted RGB features and depth cues to infer a saliency map. Due to their lack of reliance on manual human annotation, these traditional methods can be regarded as early manifestations of unsupervised SOD. Ju *et al*. (Ju et al., 2014) consider the incorporation of a prior induced from anisotropic center-surround operator, and an additional depth prior. Ren *et al*. (Ren et al., 2015) also introduce two priors: normalized depth prior and the global-context surface orientation prior. In contrary to direct utilization of the depth contrast priors, local background enclosure prior is explicitly developed by (Feng et al., 2016). Interested readers may refer to (Fan et al., 2020a; Peng et al., 2014; Zhou et al., 2021) for more in-depth surveys and analyses. However, by relying on manually-crafted priors, these methods tend to have inferior performance.

Meanwhile, considerable performance gain has been achieved by recent efforts in RGB-based unsupervised SOD (Zhang et al., 2017; 2018; Nguyen et al., 2019; Hsu et al., 2018a), which instead construct automated feature representations using deep learning. A typical strategy is to leverage the noisy output produced by traditional methods as pseudo-label (*i.e.*, supervisory signal) for training saliency prediction net. The pioneering work of Zhang *et al*. (Zhang et al., 2017) fuses the outputs of multiple unsupervised saliency models as guiding signals in CNN training. In (Zhang et al., 2018), competitive performance is achieved by fitting a noise modeling module to the noise distribution of pseudo-label. Instead of directly using pseudo-labels from handcrafted methods, Nguyen *et al*. (Nguyen et al., 2019) further refine pseudo-labels via a self-supervision iterative process. Besides, deep unsupervised learning has been considered by (Hsu et al., 2018b; Tsai et al., 2019) for the co-saliency task, where superior performance has been obtained by engaging a fusion-learning scheme (Hsu et al., 2018b), or utilizing dedicated loss terms (Tsai et al., 2019). These methods demonstrate that the incorporation of powerful deep neural network brings better feature representation than those unsupervised SOD counterparts based on handcrafted features.

In this work, a principled research investigation on *deep unsupervised RGB-D saliency detection*. is presented for the first time. Different from existing deep RGB-based unsupervised SOD (Zhang et al., 2017; 2018) that use pseudo-labels from the handcrafted methods directly, our work aims to refine and update pseudo-labels to remove the undesired noise. And compared to the refinement method (Nguyen et al., 2019) using self-supervision technique, our approach instead adopts disentangled depth information to promote pseudo-labels.

## 3 METHODOLOGY

### 3.1 OVERVIEW

Fig. 2 presents an overview of our Depth-disentangled Saliency Update (DSU) framework. Overall our DSU strives to significantly improve the quality of pseudo-labels, that leads to more trustworthy supervision signals for training the saliency network. It consists of three key components. **First** is a saliency network responsible for saliency prediction, whose initial supervision signal is provided by traditional handcrafted method without using human annotations. **Second**, a depth network and a depth-disentangled network are designed to decompose depth cues into saliency-guided depth $D_{Sal}$ and non-saliency-guided depth $D_{NonSal}$, to explicitly depict saliency cues in the spatial layout. **Third**, a depth-disentangled label update (DLU) module is devised to refine and update the pseudo-labels, by engaging the learned $D_{Sal}$ and $D_{NonSal}$. The updated pseudo-labels could in turn provide more trustworthy supervisions for the saliency network. Moreover, an attentive training strategy (ATS) is incorporated when training the saliency network, tailored for noisy unsupervised learning by mitigating the ambiguities caused by noisy pseudo-labels. We note that the DSU and ATS are not performed at test time, so does not affect the inference speed. In other words, the inference stage

---

In this paper, unsupervised learning refers to learning without using human annotation (Zhang et al., 2017).

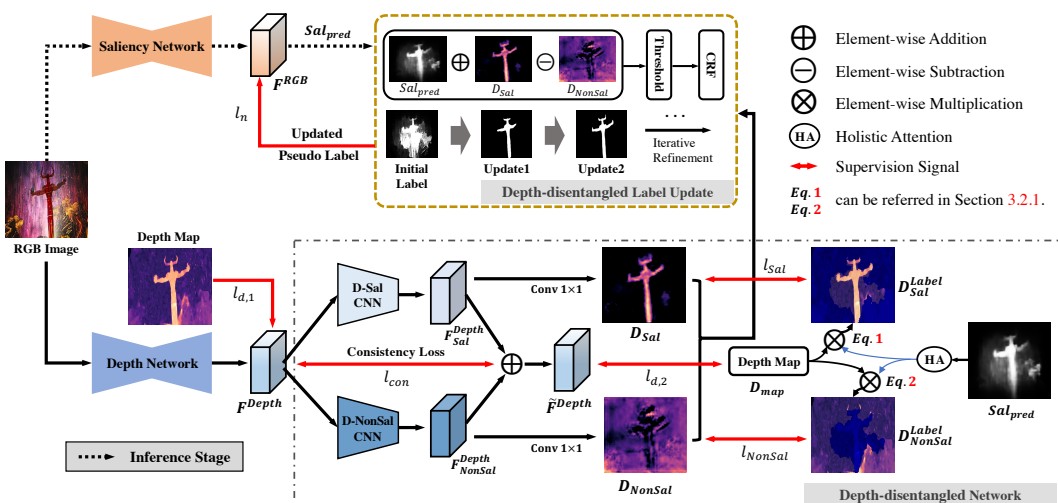

Figure 2: Overview of the proposed method. The saliency network is trained with the iteratively updated pseudo-labels. The depth network and depth-disentangled network are designed to decompose raw depth into saliency-guided depth $D_{Sal}$ and non-saliency-guided depth $D_{NonSal}$, which are subsequently fed into the depth-disentangled label update (DLU) module to refine and update pseudo-labels. The inference stage involves only the black dashed portion.

involves only the black dashed portion of the proposed network architecture in Fig. 2, *i.e.*, only RGB images are used for predicting saliency, which enables very efficient detection.

## 3.2 DEPTH-DISENTANGLED SALIENCY UPDATE

The scene geometry embedded in the depth map can serve as a complementary information source to refine the pseudo-labels. Nevertheless, as discussed in the introduction section, a direct adoption of the raw depth may not necessarily lead to good performance due to the large value variations and inherit noises in raw depth map. Without explicit pixel-wise supervision, the saliency model may unfortunately be confused by nearby background regions having the same depth but with different contexts; meanwhile, depth variations in salient or background regions may also result in false responses. These observations motivate us to design a depth-disentangled network to effectively capture discriminative saliency cues from depth, and a depth-disentangled label update (DLU) strategy to improve and refine pseudo-labels by engaging the disentangled depth knowledges.

### 3.2.1 DEPTH-DISENTANGLED NETWORK

The depth-disentangled network aims at capturing valuable saliency as well as redundant non-salient cues from raw depth map. As presented in the bottom of Fig. 2, informative depth feature $F^{Depth}$ is first extracted from the depth network under the supervision of raw depth map, using the mean square error (MSE) loss function, *i.e.*, $l_{d,1}$. The $F^{Depth}$ is then decomposed into saliency-guided depth $D_{Sal}$ and non-saliency-guided depth $D_{NonSal}$ following two principles: 1) explicitly guiding the model to learn saliency-specific cues from depth; 2) ensuring the coherence between the disentangled and original depth features.

Specifically, in the bottom right of Fig. 2, we first construct the spatial supervision signals for the depth-disentangled network. Given the rough saliency prediction $Sal_{pred}$ from the saliency network and the raw depth map $D_{map}$, the (non-)saliency-guided depth masks, *i.e.*, $D_{Sal}^{Label}$ and $D_{NonSal}^{Label}$, can be obtained by multiplying $Sal_{pred}$ (or $1 - Sal_{pred}$) and depth map $D_{map}$ in a spatial attention manner. Since the predicted saliency may contain errors introduced from the inaccurate pseudo-labels, we employ a holistic attention (HA) operation (Wu et al., 2019) to smooth the coverage area of the predicted saliency, so as to effectively perceive more saliency area from depth. Formally, the (non-)saliency-guided depth masks are generated by:

$$D_{Sal}^{Label} = \Psi_{\max}(\mathcal{F}_G(Sal_{pred}, k), Sal_{pred}) \otimes D_{map}, \quad (1)$$

$$D_{NonSal}^{Label} = \Psi_{\max}(\mathcal{F}_G(1 - Sal_{pred}, k), 1 - Sal_{pred}) \otimes D_{map}, \quad (2)$$

where $\mathcal{F}_G(\cdot, k)$ represents the HA operation, which is implemented using the convolution operation with Gaussian kernel $k$ and zero bias; the size and standard deviation of the Gaussian kernel $k$ are initialized with 32 and 4, respectively, which are then finetuned through the training procedure;

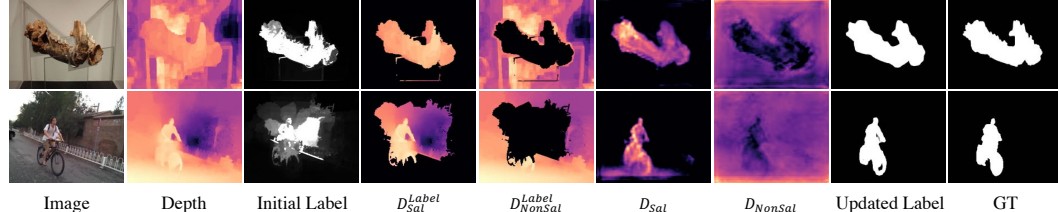

Image   Depth   Initial Label   $D_{Sal}^{Label}$   $D_{NonSal}^{Label}$   $D_{Sal}$   $D_{NonSal}$   Updated Label   GT

Figure 3: The internal inspections of the proposed DSU. It is observed that the updated label exhibits more reliable saliency signals than initial pseudo-label.

$\Psi_{\max}(\cdot, \cdot)$ is a maximum function to preserve the higher values from the Gaussian filtered map and the original map; $\otimes$ denotes pixel-wise multiplication.

Building upon the guidance of $D_{Sal}^{Label}$ and $D_{NonSal}^{Label}$, $F^{Depth}$ is fed into D-Sal CNN and D-NonSal CNN to explicitly learn valuable saliency and redundant non-salient cues from depth map, generating $D_{Sal}$ and $D_{NonSal}$, respectively. The loss functions here (*i.e.*, $l_{Sal}$ and $l_{NonSal}$) are MSE loss. Detailed structures of D-Sal and D-NonSal CNNs are in the appendix. To further ensure the coherence between the disentangled and original depth features, a consistency loss, $l_{con}$, is employed as:

$$l_{con} = \frac{1}{H \times W \times C} \sum_{i=1}^{H} \sum_{j=1}^{W} \sum_{k=1}^{C} \|F_{i,j,k}^{Depth}, \tilde{F}_{i,j,k}^{Depth}\|_2, \tag{3}$$

where $\tilde{F}^{Depth}$ is the sum of the disentangled $F_{Sal}^{Depth}$ and $F_{NonSal}^{Depth}$, which denotes the regenerated depth feature; $H$, $W$ and $C$ are the height, width and channel of $F^{Depth}$ and $\tilde{F}^{Depth}$; $\|\cdot\|_2$ represents Euclidean norm. Here, $\tilde{F}^{Depth}$ is also under the supervision of depth map, using MSE loss, *i.e.*, $l_{d,2}$.

Then, the overall training objective for the depth network and the depth-disentangled network is as:

$$\mathcal{L}_{depth} = \frac{1}{5N} \sum_{n=1}^{N} (l_{d,1}^n + l_{d,2}^n + l_{Sal}^n + l_{NonSal}^n + \lambda l_{con}^n), \tag{4}$$

where $n$ denotes the $n_{th}$ sample in a mini-batch with $N$ training samples; $\lambda$ is set to 0.02 in the experiments to balance the consistency loss $l_{con}$ and other loss terms. As empirical evidences in Fig. 3 suggested, comparing to the raw depth map, $D_{Sal}$ is better at capturing discriminative cues of the salient objects, while $D_{NonSal}$ better depicts the complementary background context.

In the next step, the learned $D_{Sal}$ and $D_{NonSal}$ are fed into our Depth-disentangled Label Update, to obtain the improved pseudo-labels.

### 3.2.2 Depth-disentangled Label Update

To maintain more reliable supervision signals in training, a depth-disentangled label update (DLU) strategy is devised to iteratively refine & update pseudo-labels. Specifically, as shown in the upper stream of Fig. 2, using the obtained $Sal_{pred}$, $D_{Sal}$ and $D_{NonSal}$, the DLU simultaneously highlights the salient regions in the coarse saliency prediction by the sum of $Sal_{pred}$ and $D_{Sal}$, and suppresses the non-saliency negative responses by subtracting $D_{NonSal}$ in a pixel-wise manner. This process can be formulated as:

$$\mathcal{S}_{temp} = Sal_{pred}^{i,j} + D_{Sal}^{i,j} - D_{NonSal}^{i,j}\Big|_{i \in [1,H]; j \in [1,W]}. \tag{5}$$

To avoid the value overflow of the obtained $\mathcal{S}_{temp}$ (*i.e.*, removing negative numbers and normalizing the results to the range of [0, 1]), a thresholding operation and a normalization process are performed as:

$$\mathcal{S}_{\mathcal{N}} = \frac{\mathcal{S}_n^{i,j} - min(\mathcal{S}_n)}{max(\mathcal{S}_n) - min(\mathcal{S}_n)}, \text{where } \mathcal{S}_n = \begin{cases} 0, & \text{if } \mathcal{S}_{temp}^{i,j} < 0 \\ \mathcal{S}_{temp}^{i,j}, & \text{others} \end{cases}, i \in [1, H]; j \in [1, W], \tag{6}$$

where $min(\cdot)$ and $max(\cdot)$ denote the minimum and maximum functions. Finally, a fully-connected conditional random field (CRF (Hou et al., 2017)) is applied to $\mathcal{S}_{\mathcal{N}}$, to generate the enhanced saliency map $\mathcal{S}_{map}$ as the updated pseudo-labels. The empirical result of the proposed DSU step is showcased in Fig. 3, with quantitative results presented in Table 4. These internal evidences suggest that the DSU leads to noticeable improvement after merely several iterations.

### 3.3 ATTENTIVE TRAINING STRATEGY

When training the saliency network using pseudo-labels, an attentive training strategy (ATS) is proposed to tailor for the deep unsupervised learning context, to reduce the influence of ambiguous pseudo-labels, and concentrate on the more reliable training examples. This strategy is inspired by the human learning process of understanding new knowledge, that is, from general to specific understanding cycle (Dixon, 1999; Peltier et al., 2005). The ATS alternates between two steps to re-weigh the training instances in a mini-batch.

To be specific, we first start by settling the related loss functions. For the $n_{th}$ sample in a mini-batch with $N$ training samples, we define the binary cross-entropy loss between the predicted saliency $Sal_{pred}^n$ and the pseudo-label $\mathcal{S}_{map}^n$ as:

$$l_n = -(\mathcal{S}_{map}^n \cdot \log Sal_{pred}^n + (1 - \mathcal{S}_{map}^n) \cdot \log(1 - Sal_{pred}^n)). \tag{7}$$

Then, the training objective for the saliency network in current mini-batch is defined as an attentive binary cross-entropy loss $\mathcal{L}_{sal}$, which can be represented as follows:

$$\mathcal{L}_{sal} = \frac{1}{\sum_{n=1}^{N} \alpha_n} \sum_{n=1}^{N} (\alpha_n \cdot l_n), \alpha_n = \begin{cases} 1, & \text{step one,} \\ \frac{\sum_{i \in N}^{i \neq n} e^{l_i}}{\sum_{i \in N} e^{l_i}}, & \text{step two,} \end{cases} \tag{8}$$

where $\alpha_n$ represents the weight of the $n_{th}$ training sample at current training mini-batch.

The ATS starts from a uniform weight for each training sample in step one to learn general representations across a lot of training data. Step two decreases the importance of ambiguous training instances through the imposed attentive loss; the higher the loss value, the less weight an instance is to get.

In this paper, we define step one and two as a training round ($2\tau$ epochs). During each training round, the saliency loss $\mathcal{L}_{sal}$ and depth loss $\mathcal{L}_{depth}$ are optimized simultaneously to train their network parameters. The proposed DLU is taken at the end of each training round to update the pseudo-labels for the saliency network; meanwhile, $D_{Sal}^{Label}$ and $D_{NonSal}^{Label}$ in Eqs.1 and

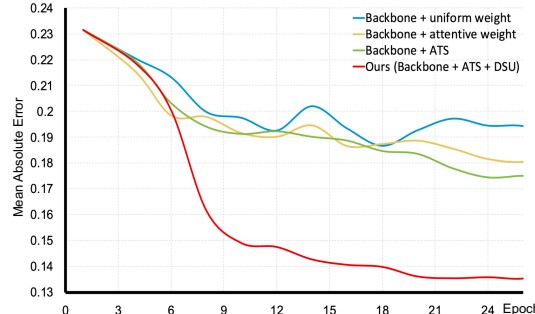

Figure 4: Analysis of the proposed attentive training strategy, when evaluated on NJUD testset. The 'backbone' refers to the saliency network trained with initial pseudo-labels.

2 are also updated using the improved $Sal_{pred}$. This allows the network to make efficient use of the updated pseudo labels. As suggested by Fig. 4 and Table 3, the use of our attentive training strategy leads to more significant error reduction compared with the backbone solely using uniform weight or attentive weight for each instance. Besides, when our DSU and the attentive training strategy are both incorporated to the backbone, better results are achieved.

## 4 EXPERIMENTS AND ANALYSES

### 4.1 DATASETS AND EVALUATION METRICS

Extensive experiments are conducted over four large-scale RGB-D SOD benchmarks. NJUD (Ju et al., 2014) in its latest version consists of 1,985 samples, that are collected from the Internet and 3D movies; NLPR (Peng et al., 2014) has 1,000 stereo images collected with Microsoft Kinect; STERE (Niu et al., 2012) contains 1,000 pairs of binocular images downloaded from the Internet; DUTLF-Depth (Piao et al., 2019) has 1,200 real scene images captured by a Lytro2 camera. We follow the setup of (Fan et al., 2020a) to construct the training set, which includes 1,485 samples from NJUD and 700 samples from NLPR, respectively. Data augmentation is also performed by randomly rotating, cropping and flipping the training images to avoid potential overfitting. The remaining images are reserved for testing.

Here, five widely-used evaluation metrics are adopted: E-measure ($E_\xi$) (Fan et al., 2018), weighed F-measure ($F_\beta^w$) (Margolin et al., 2014), F-measure ($F_\beta$) (Achanta et al., 2009), Mean Absolute Error (MAE or $\mathcal{M}$) (Borji et al., 2015), and inference time(s) or FPS (Frames Per Second).

Implementation details are presented in the Appx A.3. *Source code is publicly available.*

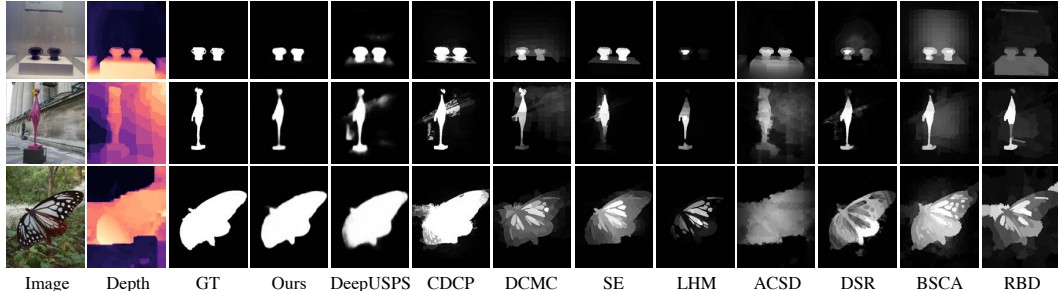

| Image | Depth | GT | Ours | DeepUSPS | CDCP | DCMC | SE | LHM | ACSD | DSR | BSCA | RBD |

Figure 5: Qualitative comparison with unsupervised saliency detection methods. GT denotes ground-truth.

Table 1: Quantitative comparison with unsupervised SOD methods. 'Backbone' refers to the saliency feature extraction network (Wu et al., 2019) adopted in our pipeline, *i.e.* the one without the two proposed key components. The RGB-based methods are specifically marked by †. UnSOD is shorthand for unsupervised SOD. We also provide the results of existing fully supervised methods that can be referenced in Table 8.

| | * | Inference Time(s)↓ | NJUD $E_\xi$↑ | $F_\beta^w$↑ | $F_\beta$↑ | $\mathcal{M}$↓ | NLPR $E_\xi$↑ | $F_\beta^w$↑ | $F_\beta$↑ | $\mathcal{M}$↓ | STERE $E_\xi$↑ | $F_\beta^w$↑ | $F_\beta$↑ | $\mathcal{M}$↓ | DUTLF-Depth $E_\xi$↑ | $F_\beta^w$↑ | $F_\beta$↑ | $\mathcal{M}$↓ |
|---|---|---|---|---|---|---|---|---|---|---|---|---|---|---|---|---|---|---|
| Handcrafted UnSOD | RBD† (Zhu et al., 2014) | 0.189 | .684 | .387 | .556 | .256 | .765 | .388 | .590 | .211 | .730 | .443 | .610 | .223 | .733 | .447 | .619 | .222 |
| | MST† (Tu et al., 2016) | 0.030 | .670 | .291 | .436 | .281 | .762 | .257 | .491 | .199 | .681 | .312 | .447 | .269 | .678 | .254 | .401 | .279 |
| | BSCA† (Qin et al., 2015) | 2.665 | .756 | .446 | .623 | .216 | .745 | .376 | .554 | .178 | .803 | .497 | .676 | .179 | .808 | .479 | .682 | .181 |
| | DSR† (Li et al., 2013) | 0.376 | .739 | .436 | .594 | .196 | .757 | .451 | .545 | .120 | .785 | .486 | .645 | .165 | .797 | .478 | .640 | .164 |
| | ACSD (Ju et al., 2014) | 0.718 | .790 | .448 | .696 | .198 | .751 | .327 | .547 | .171 | .793 | .425 | .661 | .200 | .250 | .210 | .188 | .668 |
| | DES (Cheng et al., 2014) | 7.790 | .421 | .241 | .165 | .448 | .735 | .259 | .583 | .301 | .673 | .383 | .592 | .297 | .733 | .386 | .668 | .280 |
| | LHM (Peng et al., 2014) | 2.130 | .722 | .311 | .625 | .201 | .772 | .320 | .520 | .119 | .772 | .360 | .703 | .171 | .767 | .350 | .659 | .174 |
| | GP (Ren et al., 2015) | 12.98 | .730 | .323 | .666 | .204 | .813 | .347 | .670 | .144 | .785 | .371 | .710 | .182 | - | - | - | - |
| | CDB (Liang et al., 2018) | 0.600 | .752 | .408 | .650 | .200 | .810 | .388 | .618 | .108 | .808 | .436 | .713 | .166 | - | - | - | - |
| | SE (Guo et al., 2016) | 1.570 | .780 | .518 | .735 | .164 | .853 | .578 | .701 | .085 | .825 | .546 | .747 | .143 | .730 | .339 | .474 | .196 |
| | DCMC (Cong et al., 2016) | 1.210 | .796 | .506 | .715 | .167 | .684 | .265 | .328 | .196 | .832 | .529 | .743 | .148 | .712 | .290 | .406 | .243 |
| | MB (Zhu et al., 2017a) | - | .643 | .369 | .492 | .202 | .814 | .574 | .637 | .089 | .693 | .455 | .572 | .178 | .691 | .464 | .577 | .156 |
| | CDCP (Zhu et al., 2017b) | 5.720 | .751 | .522 | .618 | .181 | .785 | .512 | .591 | .114 | .797 | .596 | .666 | .149 | .794 | .530 | .633 | .159 |
| Deep UnSOD | USD† (Zhang et al., 2018) | 0.0180 | .768 | .565 | .630 | .163 | .786 | .536 | .580 | .119 | .796 | .572 | .670 | .146 | .795 | .545 | .650 | .157 |
| | DeepUSPS† (Nguyen et al., 2019) | 0.0292 | .771 | .576 | .647 | .159 | .809 | .622 | .639 | .088 | .806 | .632 | .682 | .124 | .798 | .573 | .654 | .149 |
| | Backbone | 0.0286 | .759 | .510 | .627 | .186 | .760 | .479 | .570 | .126 | .794 | .555 | .666 | .158 | .798 | .512 | .644 | .167 |
| | Δ gains | - | ↑5% | ↑17% | ↑15% | ↓27% | ↑16% | ↑37% | ↑31% | ↓48% | ↑8% | ↑22% | ↑16% | ↓37% | ↑7% | ↑27% | ↑18% | ↓36% |
| | **Ours** | 0.0286 | **.797** | **.597** | **.719** | **.135** | **.879** | **.657** | **.745** | **.065** | **.857** | **.678** | **.774** | **.099** | **.854** | **.650** | **.763** | **.107** |

## 4.2 COMPARISON WITH THE STATE-OF-THE-ARTS

Our approach is compared with 15 unsupervised SOD methods, *i.e.*, without using any human annotations. Their results are either directly furnished by the authors of the respective papers, or generated by re-running their original implementations. In this paper, we make the first attempt to address deep-learning-based unsupervised RGB-D SOD. Since existing unsupervised RGB-D methods are all based on handcrafted feature representations, we additionally provide several RGB-based methods (*e.g.*, USD and DeepUSPS) for reference purpose only. This gives more observational evidences for the related works. These RGB-based methods are specifically marked by † in Table 1.

Quantitative results are listed in Table 1, where our approach clearly outperforms the state-of-the-art unsupervised SOD methods in both RGB-D and RGB only scenarios. This is due to our DSU framework that leads to trustworthy supervision signals for saliency network. Furthermore, our network design leads to a light-weight architecture in the inference stage, shown as the black dashed portion in Fig. 2. This enables efficient & effective detection of salient objects and brings a large-margin improvement over the backbone network without introducing additional depth input and computational costs, as shown in Fig. 1 and Table 1. Qualitatively, saliency predictions of competing methods are exhibited in Fig. 5. These results consistently proves the superiority of our method.

## 4.3 ANALYSIS OF THE EMPIRICAL RESULTS

The focus here is on the evaluation of the contributions from each of the components, and the evaluation of the obtained pseudo-labels as intermediate results.

**Effect of each component.** In Table 2, we conduct ablation study to investigate the contribution of each component. To start with, we consider the backbone (a), where the saliency network is trained with initial pseudo-labels. As our proposed ATS and DSU are gradually incorporated, increased performance has been observed on both datasets. Here we first investigate the benefits of ATS by applying it to the backbone and obtaining (b). We observe increased F-measure scores of 3% and 5.8% on NJUD and NLPR benchmarks, respectively. This clearly shows that the proposed ATS can effectively improve the utilization of reliable pseudo-labels, by re-weighting ambiguous training data. We then investigate in detail all components in our DSU strategy. The addition of the entire DSU leads to (f), which significantly improves the F-measure metric by 11.3% and 23.5% on each

of the benchmarks, while reducing the MAE by 22.4% and 41.9%, respectively. This verifies the effectiveness of the DSU strategy to refine and update pseudo-labels. Moreover, as we gradually exclude the consistency loss $l_{con}$ (row (e)) and HA operation (row (d)), degraded performances are observed on both datasets. For an extreme case where we remove the DLU and only maintain CRF to refine pseudo-labels, it is observed that much worse performance is achieved. These results consistently demonstrate that all components in the DSU strategy are beneficial for generating more accurate pseudo-labels.

We also display the visual evidence of the updated pseudo-labels obtained from the DSU strategy in Fig. 6. It is shown that the initial pseudo-labels unfortunately tend to miss important parts as well as fine-grained details. The application of CRF helps to filter away background

Table 2: Ablation study of our deep unsupervised RGB-D SOD pipeline, using the F-measure and MAE metrics.

| Index | Model Setups | | NJUD | | NLPR | |
|---|---|---|---|---|---|---|
| | | | $F_\beta \uparrow$ | $\mathcal{M} \downarrow$ | $F_\beta \uparrow$ | $\mathcal{M} \downarrow$ |
| (a) | | Backbone | 0.627 | 0.186 | 0.570 | 0.126 |
| (b) | | (a) + attentive training strategy | 0.646 | 0.174 | 0.603 | 0.112 |
| (c) | | (b) + CRF | 0.674 | 0.160 | 0.663 | 0.093 |
| (d) | DSU strategy | (b) + DSU (w/o $l_{con}$&$HA$) | 0.703 | 0.141 | 0.716 | 0.074 |
| (e) | | (b) + DSU (w/o $l_{con}$) | 0.712 | 0.137 | 0.735 | 0.068 |
| (f) | | (b) + DSU (**Ours**) | 0.719 | 0.135 | 0.745 | 0.065 |

noises, while salient parts could still be missing out. By adopting our DSU and attentive training strategy, the missing parts could be retrieved in the updated pseudo-labels, with the object silhouette also being refined. These numerical and visual results consistently verify the effectiveness of our pipeline in deep unsupervised RGB-D saliency detection.

**Detailed analysis of the ATS.** The influence of the attentive training strategy (ATS) is studied in detail here. The error reduction curves in Fig. 4 have shown that the use of ATS could lead to greater test error reduction compared with the backbone that uses uniform weight or attentive weight for each instance.

Table 3: Analyzing attentive training strategy (ATS) with different settings. 'Setting 1' is backbone + uniform weight + DSU, and 'Setting 2' is backbone + attentive weight + DSU. The last four columns show backbone + ATS + DSU with different alternation interval $\tau$.

| * | | Analysis of the ATS | | **Ours** | Analysis of the interval $\tau$ | | |
|---|---|---|---|---|---|---|---|
| | | Setting 1 | Setting 2 | $\tau = 3$ | $\tau = 1$ | $\tau = 5$ | $\tau = 10$ |
| NJUD | $F_\beta \uparrow$ | 0.695 | 0.707 | **0.719** | 0.711 | 0.712 | 0.687 |
| | $\mathcal{M} \downarrow$ | 0.148 | 0.142 | **0.135** | 0.143 | 0.138 | 0.147 |
| NLPR | $F_\beta \uparrow$ | 0.706 | 0.737 | **0.745** | 0.742 | 0.739 | 0.698 |
| | $\mathcal{M} \downarrow$ | 0.071 | 0.067 | **0.065** | 0.069 | 0.066 | 0.073 |

When combining the ATS with the DSU strategy as shown in Table 3, the ATS also enables the network to learn more efficiently than using uniform weight or attentive weight alone, by comparing 'Ours' with 'Setting 1' and 'Setting 2'. In addition, we discuss the effect of different alternation intervals $\tau$ in ATS. As listed in Table 3, the larger or smaller interval leads to inferior performance due to the insufficient or excessive learning of saliency models.

Table 4: Internal mean absolute errors, each is evaluated between current pseudo-labels and the corresponding true labels (only used for evaluation purpose) during the training process.

| Pseudo-label Update | Initial | Update 1 | Update 2 | Update 3 | Update 4 |
|---|---|---|---|---|---|
| Mean absolute error | 0.162 | 0.124 | 0.117 | 0.116 | 0.116 |

Table 5: Comparison of different pseudo-label generation variants. 'CRF' refers to fully-connected CRF. 'OTSU' represents the standard Otsu image thresholding method.

| Label Accuray | $E_\xi \uparrow$ | $F_\beta^w \uparrow$ | $F_\beta \uparrow$ | $\mathcal{M} \downarrow$ |
|---|---|---|---|---|
| Initial pseudo-label | 0.760 | 0.526 | 0.614 | 0.162 |
| Initial pseudo-label + CRF | 0.763 | 0.578 | 0.634 | 0.144 |
| Our DSU | **0.792** | **0.635** | **0.708** | **0.116** |
| Depth map | 0.419 | 0.284 | 0.164 | 0.414 |
| Depth map + OTSU | 0.465 | 0.398 | 0.429 | 0.332 |

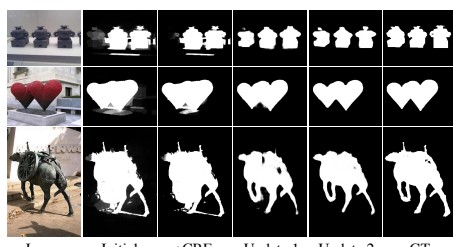

Image   Initial   +CRF   Update 1   Update 2   GT

Figure 6: Visual examples of the intermediate pseudo-labels used in our approach. 'Initial' shows the initial pseudo-labels generated by traditional handcrafted method. '+CRF' refers to the pseudo-labels after applying fully-connected CRF. Update 1&2 represent the updated pseudo-labels produced in our pipeline over two training rounds. 'GT' means the ground truth, used for reference purpose only.

**Analysis of pseudo-labels.** We analyze the quality of pseudo-labels over the training process in Table 4, where the mean absolute error scores between the pseudo-labels at different update rounds and the ground-truth labels are reported. It is observed that the quality of pseudo-labels is significantly improved during the first two rounds, which then remains stable in the consecutive rounds. Fig. 6 also shows that the initial pseudo-label is effectively refined, where the updated pseudo-label is close to the true label. This provides more reliable guiding signals for training the saliency network.

**Comparison of different pseudo-label variants.** In Table 5, we investigate other possible pseudo-label generation variants, including, initial pseudo-label with CRF refinement, raw depth map, and raw depth map together with OTSU thresholding (Otsu, 1979). It is shown that, compared with the direct use of CRF, our proposed DSU is able to provide more reliable pseudo-labels, by disentangling depth to promote saliency. It is worth noting that a direct application of the raw depth map or together with an OTSU adaptive thresholding of the depth map, may nevertheless lead to awful results. We conjecture this is because of the large variations embedded in raw depth, and the fact that foreground objects may be affected by nearby background stuffs that are close in depth.

**Is depth-disentanglement necessary?** In our DSU, we disentangle the raw depth map into valuable salient parts as well as redundant non-salient areas to simultaneously highlight the salient regions and suppress the non-salient negative responses. To verify the necessity

Table 6: Discussion on different pseudo-label updating settings.

| Different Pseudo-label Updating Setups | NJUD | | NLPR | |
|---|---|---|---|---|
| | $F_\beta \uparrow$ | $\mathcal{M} \downarrow$ | $F_\beta \uparrow$ | $\mathcal{M} \downarrow$ |
| (a) DSU w/o $D_{NonSal}$ | 0.698 | 0.143 | 0.715 | 0.072 |
| (b) DSU using $D_{Sal}^{Label}$, $D_{NonSal}^{Label}$ | 0.691 | 0.154 | 0.707 | 0.079 |
| (c) DSU using $D_{Sal}$, $D_{NonSal}$ (**Ours**) | 0.719 | 0.135 | 0.745 | 0.065 |

of depth disentanglement, we remove the $D_{NonSal}$ branch in the depth-disentangled network and only use the $D_{Sal}$ to update pseudo-labels in DLU. Empirical results in Table 6 reveal that removing $D_{NonSal}$ (row (a)) leads to significant inferior performance when comparing with the original DSU (row (c)), which proves the superiority of our DSU design.

**How about using raw depth labels in DSU?** We also consider to directly utilize the raw disentangled depth labels, *i.e.,* $D_{Sal}^{Label}$ and $D_{NonSal}^{Label}$, to update pseudo-labels in DSU. However, a direct adoption of the raw depth labels does not lead to good performance, as empirically exhibited in Table 6 (b). The DSU using raw labels performs worse compared to our original DSU design in row (c). This is partly due to the large value variations and intrinsic label noises in raw depth map as discussed before. On the other hand, benefiting from the capability of deep-learning-based networks to learn general representations from large-scale data collections, using $D_{Sal}$ and $D_{NonSal}$ in our DSU strategy is able to eliminate the potential biases in a single raw depth label. Visual evidences as exhibited in the $4^{th}$-$7^{th}$ columns of Fig. 3 also support our claim.

### 4.4 APPLICATION TO FULLY-SUPERVISED SETTING

To show the generic applicability of our approach, a variant of our DSU is applied to several cutting-edge fully-supervised SOD models to improve their performance. This is made possible by re-defining the quantities $D_{Sal}^{Label} = \mathcal{S}_{GT} \otimes D_{map}$ and $D_{NonSal}^{Label} = (1 - \mathcal{S}_{GT}) \otimes D_{map}$, with $\mathcal{S}_{GT}$ being the ground-truth saliency. Then the saliency network (*i.e.,* existing SOD models) and the depth-disentangled network are retrained by $\mathcal{S}_{GT}$ and the new $D_{Sal}^{Label}$ and $D_{NonSal}^{Label}$, respectively. After training, the proposed DLU is engaged to obtain the final improved saliency. In Table 7, we report

Table 7: Applying our DSU to existing fully-supervised RGB-D SOD methods.

| * | NJUD | | NLPR | |
|---|---|---|---|---|
| | $F_\beta \uparrow$ | $\mathcal{M} \downarrow$ | $F_\beta \uparrow$ | $\mathcal{M} \downarrow$ |
| DMRA (Piao et al., 2019) | 0.872 | 0.051 | 0.855 | 0.031 |
| **+ Our DSU** | 0.893 | 0.044 | 0.879 | 0.026 |
| CMWN (Li et al., 2020c) | 0.878 | 0.047 | 0.859 | 0.029 |
| **+ Our DSU** | 0.901 | 0.041 | 0.882 | 0.025 |
| FRDT (Zhang et al., 2020f) | 0.879 | 0.048 | 0.868 | 0.029 |
| **+ Our DSU** | 0.903 | 0.038 | 0.901 | 0.023 |
| CPD (Wu et al., 2019) | 0.873 | 0.045 | 0.866 | 0.028 |
| **+ Our DSU** | 0.909 | 0.036 | 0.907 | 0.022 |

the original results of four SOD methods and the new results of incorporating our DSU strategy on two popular benchmarks. It is observed that our supervised variants have consistent performance improvement comparing to each of existing models. For example, the average MAE score of four SOD methods on NJUD benchmark is reduced by 18.0%. We attribute the performance improvement to our DSU strategy that can exploit the learned $D_{Sal}$ to facilitate the localization of salient object regions in a scene, as well as suppress the redundant background noises by subtracting $D_{NonSal}$. More experiments are presented in the Appx A.2.

## 5 CONCLUSION AND OUTLOOK

This paper tackles the new task of deep unsupervised RGB-D saliency detection. Our key insight is to internally engage and refine the pseudo-labels. This is realized by two key modules, the depth-disentangled saliency update in iteratively fine-tuning the pseudo-labels, and the attentive training strategy in addressing the issue of noisy pseudo-labels. Extensive empirical experiments demonstrate the superior performance and realtime efficiency of our approach. For future work, we plan to extend our approach to scenarios involving partial labels.

**Acknowledgement.** This research was partly supported by the University of Alberta Start-up Grant, UAHJIC Grants, and NSERC Discovery Grants (No. RGPIN-2019-04575).

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

## A  APPENDIX

In this appendix, we first elaborate on the evaluation metrics used in this paper, in Appx A.1. Then, more details and experiments of our DSU are presented in Appx A.2, including the detailed structures of D-Sal and D-NonSal CNNs, more experiments on unsupervised and fully-supervised scenarios, and other pseudo-label initialization. These empirical results consistently demonstrate the effectiveness and scalability of our approach. In Appx A.3, we illustrate on the implementation details of our unsupervised pipeline. In Appx A.4, we discuss the potential limitations which can be addressed in the near future. In Appx A.5, we additionally conduct some interesting experiments to further verify the efficiency of our method. Finally, in Appx A.6, we investigate several potential directions that deserve further exploration.

### A.1  EVALUATION METRICS

We adopt four widely-used metrics in SOD to evaluate the performance of saliency models, *i.e.*, E-measure, F-measure, weighted F-measure and MAE. *The lower the MAE, the better. For other metrics, the higher score is better.* Concretely, F-measure is an overall performance measurement and is computed by the weighted harmonic mean of the precision and recall:

$$F_\beta = \frac{(1 + \beta^2) \times Precision \times Recall}{\beta^2 \times Precision + Recall}, \tag{9}$$

where $\beta^2$ is set to 0.3 to emphasize the precision (Achanta et al., 2009). We use different fixed [0, 255] thresholds to compute the F-measure metric. Weighted F-measure ($F_\beta^w$) (Margolin et al., 2014) is its weighted measurement. MAE ($\mathcal{M}$) (Borji et al., 2015) represents the average absolute difference between the predicted saliency map and ground truth. It is used to calculate how similar a normalized saliency maps $\mathcal{S} \in [0, 1]^{W \times H}$ is compared to the ground truth $\mathcal{G} \in [0, 1]^{W \times H}$:

$$MAE = \frac{1}{W \times H} \sum_{x=1}^{W} \sum_{y=1}^{H} |\mathcal{S}(x, y) - \mathcal{G}(x, y)|, \tag{10}$$

where $W$ and $H$ denote the width and height of $\mathcal{S}$, respectively. Enhanced-alignment measure ($E_\xi$) (Fan et al., 2018) is proposed based on cognitive vision studies to capture image-level statistics and their local pixel matching information, as in

$$E_\xi = \frac{1}{W \times H} \sum_{i=1}^{W} \sum_{j=1}^{H} \phi_s(i, j), \tag{11}$$

where $\phi_s(\cdot)$ is the enhanced-alignment matrix (Fan et al., 2018), which reflects the correlation between $\mathcal{S}$ and $\mathcal{G}$ after subtracting their global means.

### A.2  MORE ANALYSES ON THE PROPOSED DSU

**Detailed structures.** The detailed structures of the D-Sal CNN and D-NonSal CNN are illustrated in Fig. 7. The 'Conv 3×3' represents a convolutional layer with the kernel size of 3.

**Visual results for unsupervised RGB-D SOD models.** More visual comparisons with unsupervised RGB-D saliency models are shown in Fig. 8, where our unsupervised pipeline generates promising saliency prediction that is close to true label.

**Comparison with current state-of-the-art RGB-D SOD methods under fully-supervised scenario.** As discussed in Sec. 4.4 of the main text, our method can boost the performance of existing SOD models greatly, by applying DSU to existing methods. In this section, we further compare our fully-supervised variant of DSU with 25 existing state-of-the-art RGB-D saliency models. We adopt the same feature-extraction backbone (Wu et al., 2019) as in DCF (Ji et al., 2021a) to build the saliency network in the DSU framework. Numerical results in Table 8 and visual results in Fig. 9 consistently show the superiority and scalability of our method.

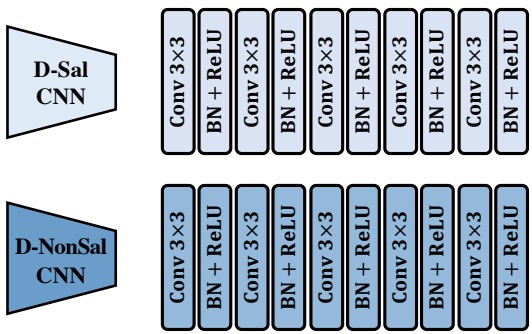

Figure 7: Detailed structure of D-Sal and D-NonSal CNNs in DSU. BN means batch normalization operation.

**Pseudo-label Initialization.** We investigate other initial pseudo-labels (*e.g.*, DCMC (Cong et al., 2016)), to further verify the effectiveness of our proposed method. The superior performance is also observed, by comparing backbone with ours: 0.167 *vs.* 0.146 on NJUD, 0.119 *vs.* 0.084 on NLPR, 0.142 *vs.* 0.119 on STERE and 0.146 *vs.* 0.128 on DUTLF-Depth, using MAE metric.

## A.3 IMPLEMENTATION DETAILS

The proposed deep unsupervised pipeline is implemented with PyTorch and trained using a single Tesla P40 GPU. Both Saliency and Depth Networks employ the same saliency feature extraction backbone (CPD (Wu et al., 2019)), which is equipped with the encoder of ResNet-50 (He et al., 2016), with initial parameters pretrained in ImageNet (Krizhevsky et al., 2012). All training & testing images are uniformly resized to the size of $352 \times 352$. Throughout training, the learning rate is set to $1 \times 10^{-4}$, and the Adam optimizer is used with a mini-batch size of 10. Our approach is trained in an unsupervised manner, *i.e.*, without any human annotations, where initial pseudo-labels are the outputs of handcrafted RGB-D saliency method CDCP (Zhu et al., 2017b). As for the proposed attentive training strategy, its alternation interval is set to 3, amounting to $2\tau = 6$ epochs in a training round. During inference, our approach directly predicts saliency maps based on an RGB image, without accessing any depth map.

## A.4 LIMITATION AND DISCUSSION

As demonstrated in this paper, our *deep unsupervised RGB-D saliency detection* approach achieves an appealing trade-off between detection accuracy and human annotation consumption. However, due to the lack of accurate pixel-level annotations, the model still fails to comprehensively detect the fine-grained details, *i.e.*, the edges of salient objects. To tackle this challenge, a doable solution is to introduce auxiliary edge constraint. For example, the edge detection loss can be employed to low-level features of the model, which forces the model to produce discriminative features highlighting object details (Zhang et al., 2020b; Liu et al., 2019). The edge maps can be generated by classical Canny operator (Canny, 1986) in an unsupervised manner. Hopefully this could encourage more inspirations and contributions to this community and further pave the way for its booming future.

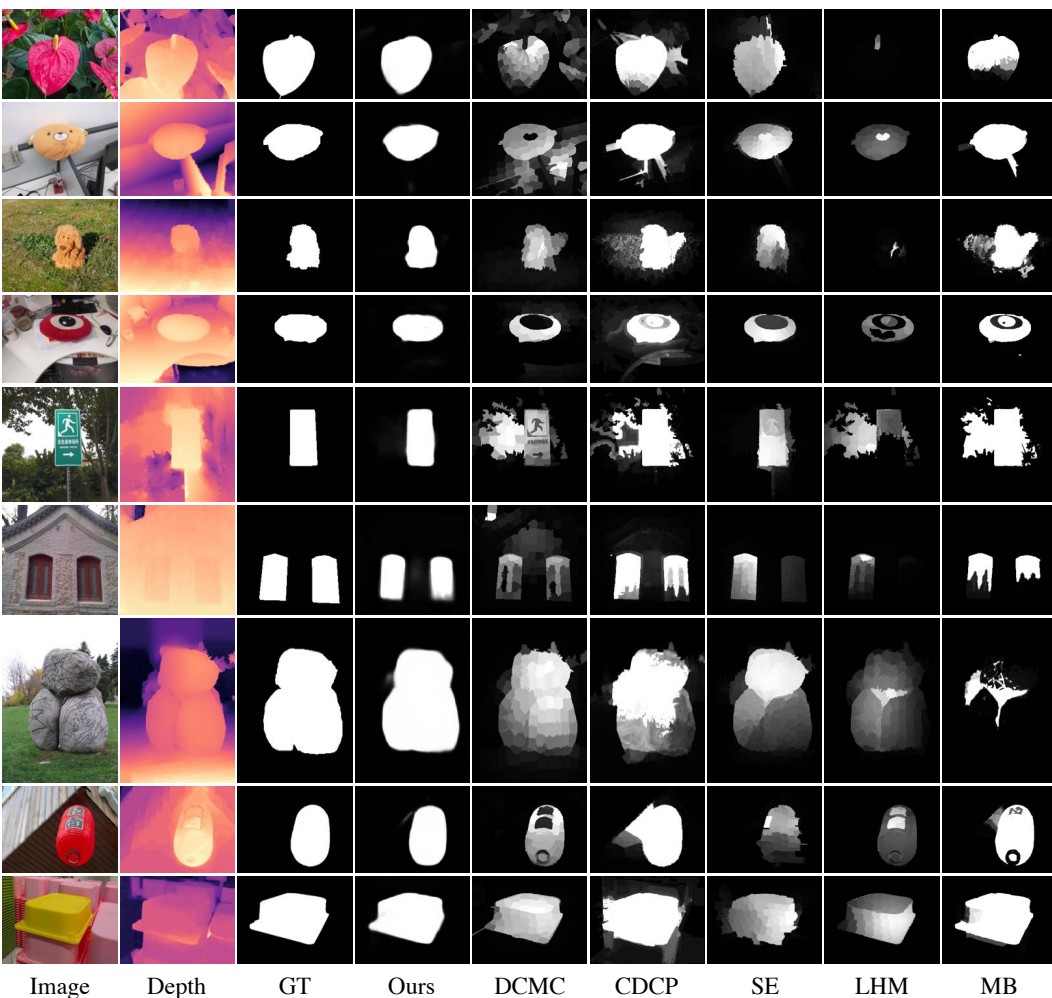

Figure 8: Qualitative comparison with unsupervised RGB-D SOD models. GT means ground-truth.

Table 8: Quantitative results of fully-supervised RGB-D saliency detection methods. The best results are highlighted in **boldface**. When evaluating the newly released DUTLF-Depth dataset, the specific setup used by (Piao et al., 2019) is adopted to make a fair comparison.

| Method | NJUD | | | | NLPR | | | | STERE | | | | DUTLF-Depth | | | |
|---|---|---|---|---|---|---|---|---|---|---|---|---|---|---|---|---|
| | $E_\xi \uparrow$ | $F_\beta^w \uparrow$ | $F_\beta \uparrow$ | $\mathcal{M} \downarrow$ | $E_\xi \uparrow$ | $F_\beta^w \uparrow$ | $F_\beta \uparrow$ | $\mathcal{M} \downarrow$ | $E_\xi \uparrow$ | $F_\beta^w \uparrow$ | $F_\beta \uparrow$ | $\mathcal{M} \downarrow$ | $E_\xi \uparrow$ | $F_\beta^w \uparrow$ | $F_\beta \uparrow$ | $\mathcal{M} \downarrow$ |
| CTMF (Han et al., 2017) | .864 | .732 | .788 | .085 | .869 | .691 | .723 | .056 | .841 | .747 | .771 | .086 | .884 | .690 | .792 | .097 |
| DF (Qu et al., 2017) | .818 | .552 | .744 | .151 | .838 | .524 | .682 | .099 | .691 | .596 | .742 | .141 | .842 | .542 | .748 | .145 |
| PCA (Chen & Li, 2018) | .896 | .811 | .844 | .059 | .916 | .772 | .794 | .044 | .887 | .801 | .826 | .064 | .858 | .696 | .760 | .100 |
| TANet (Chen & Li, 2019) | .893 | .812 | .844 | .061 | .916 | .789 | .795 | .041 | .893 | .804 | .835 | .060 | .866 | .712 | .779 | .093 |
| PDNet (Zhu et al., 2019) | .890 | .798 | .832 | .062 | .876 | .659 | .740 | .064 | .880 | .799 | .813 | .071 | .861 | .650 | .757 | .112 |
| MMCI (Chen et al., 2019) | .878 | .749 | .813 | .079 | .871 | .688 | .729 | .059 | .873 | .757 | .829 | .068 | .855 | .636 | .753 | .113 |
| CPFP (Zhao et al., 2019) | .895 | .837 | .850 | .053 | .924 | .820 | .822 | .036 | .912 | .808 | .830 | .051 | .814 | .644 | .736 | .099 |
| DMRA (Piao et al., 2019) | .908 | .853 | .872 | .051 | .942 | .845 | .855 | .031 | .923 | .841 | .876 | .049 | .927 | .858 | .883 | .048 |
| SSF (Zhang et al., 2020e) | .913 | .871 | .886 | .043 | .949 | .874 | .875 | .026 | .921 | .850 | .867 | .046 | .946 | .894 | .914 | .034 |
| A2dele (Piao et al., 2020) | .897 | .851 | .874 | .051 | .945 | .867 | .878 | .028 | .915 | .855 | .874 | .044 | .924 | .864 | .890 | .043 |
| JL-DCF (Fu et al., 2020) | - | - | - | - | .954 | .882 | .878 | **.022** | .919 | .857 | .869 | .040 | .931 | .863 | .883 | .043 |
| S2MA (Liu et al., 2020) | - | - | - | - | .938 | .852 | .853 | .030 | .907 | .825 | .855 | .051 | .921 | .861 | .866 | .044 |
| UCNet (Zhang et al., 2020a) | - | - | - | - | .953 | .878 | .890 | .025 | .922 | .867 | .885 | .039 | .903 | .821 | .856 | .056 |
| FRDT (Zhang et al., 2020f) | .917 | .862 | .879 | .048 | .946 | .863 | .868 | .029 | .925 | .858 | .872 | .042 | .941 | .878 | .902 | .039 |
| D3Net (Fan et al., 2020) | .913 | .860 | .863 | .047 | .943 | .854 | .857 | .030 | .920 | .845 | .855 | .046 | .847 | .668 | .756 | .097 |
| HDFNet (Pang et al., 2020a) | .915 | .879 | .893 | .038 | .948 | .869 | .878 | .027 | .925 | .863 | .879 | .040 | .934 | .865 | .892 | .040 |
| CMWNet (Li et al., 2020c) | .910 | .855 | .878 | .047 | .940 | .856 | .859 | .029 | .917 | .847 | .869 | .043 | .916 | .831 | .866 | .056 |
| DANet (Zhao et al., 2020c) | - | - | - | - | .949 | .858 | .871 | .028 | .914 | .830 | .858 | .047 | .925 | .847 | .884 | .047 |
| PGAR (Chen & Fu, 2020) | .915 | .871 | .893 | .042 | .955 | .881 | .885 | .024 | .919 | .856 | .880 | .041 | .944 | .889 | .914 | .035 |
| ATSA (Zhang et al., 2020c) | .921 | .883 | .893 | .040 | .945 | .867 | .876 | .028 | .919 | .866 | .874 | .040 | .947 | .901 | .918 | .032 |
| BBSNet (Fan et al., 2020b) | .924 | .884 | .902 | **.035** | .952 | .879 | .882 | .023 | .925 | .858 | .885 | .041 | .833 | .663 | .774 | .120 |
| HAINe (Li et al., 2021a) | .921 | .887 | .898 | .038 | .956 | .890 | .891 | .024 | .925 | .871 | .883 | .040 | .938 | .883 | .906 | .038 |
| RD3D (Chen et al., 2021b) | .918 | .889 | .900 | .037 | .956 | .894 | .890 | **.022** | .926 | .877 | .885 | .038 | .949 | .908 | .914 | **.031** |
| DSA$^2$F (Sun et al., 2021) | .923 | .889 | .901 | .039 | .950 | .889 | .897 | .024 | **.927** | .877 | **.895** | .038 | .949 | .908 | .924 | .032 |
| DCF (Ji et al., 2021a) | .924 | .893 | .902 | **.035** | **.957** | .892 | .891 | **.021** | **.927** | .873 | .885 | .039 | **.952** | .909 | **.926** | **.030** |
| **Ours** (fully supervised) | **.926** | **.894** | **.909** | .036 | **.957** | **.897** | **.907** | .022 | **.927** | **.882** | **.895** | .037 | .951 | **.911** | .918 | .031 |

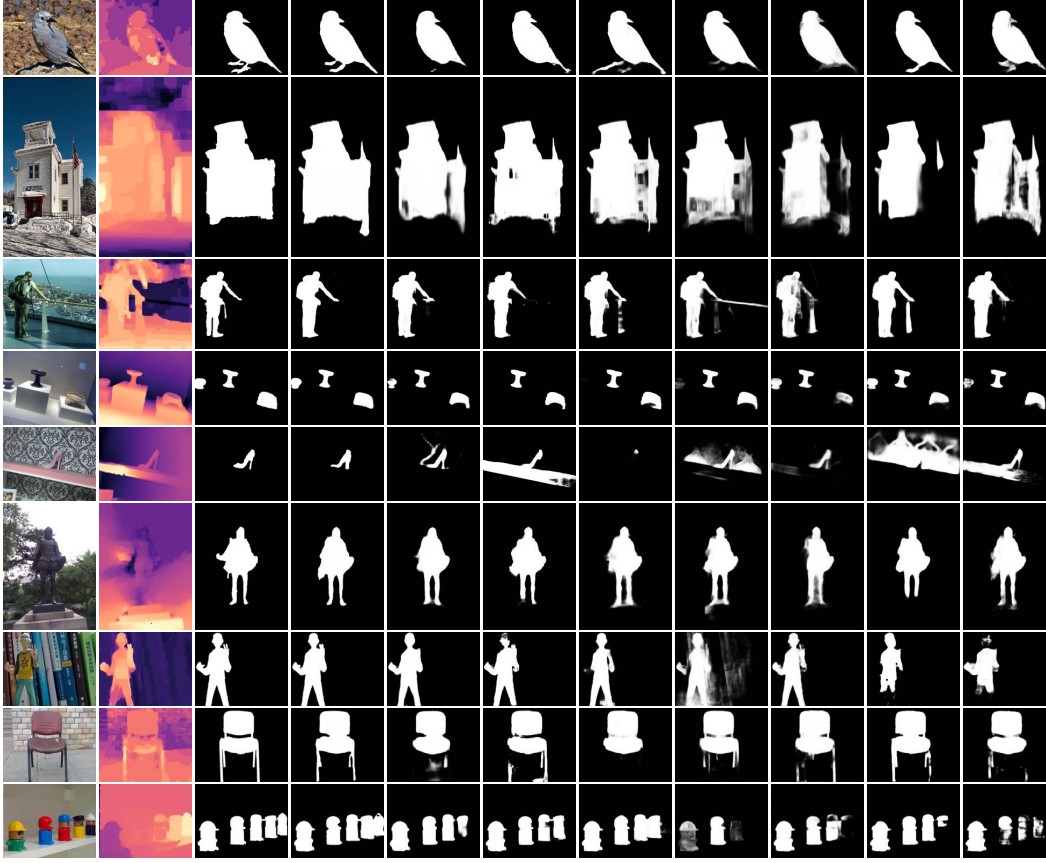

Image  Depth  GT  Ours  DCF  DSA²F  HAINet  BBSNet  S2MA  UCNet  FRDT

Figure 9: Qualitative comparison of fully-supervised RGB-D SOD methods. Obviously, our fully-supervised variant infers more appealing saliency maps compared to existing SOTA models, including DCF (Ji et al., 2021a). DSA$^2$F (Sun et al., 2021), HAINet (Li et al., 2021a), BBSNet (Fan et al., 2020b), S2MA (Liu et al., 2020), UCNet (Zhang et al., 2020a) and FRDT (Zhang et al., 2020f).

A.5    ADDITIONAL ANALYSIS

In this paper, our DSU design achieves appealing trade-off between model performance and complexity since our method has additional benefits of not introducing extra test-time cost and not relying on the depth map during the inference stage, as shown in Table 1 and Fig. 1. Meanwhile, our method can be easily adapted to integrate useful depth cues by engaging the proposed DSU and DLU. In Table 9, we conduct an interesting experiment, comparing the model performance, inference speed, FLOPs and parameter number, when using only RGB stream versus using RGB and depth simultaneously during inference. It is observed that introducing depth information can further improve the model performance, but at the same time, it will lead to the decrease of inference speed, and the increases of FLOPs and parameters. These results further demonstrate the effectiveness and efficiency of our DSU design.

Table 9: Quantitative results of our DSU when using only RGB stream versus using RGB and depth simultaneously during inference.

| | Model Complexity | | | Model Performance (**NLPR**) | | | |
| --- | --- | --- | --- | --- | --- | --- | --- |
| | Speed↑ | FLOPs↓ | Parameters↓ | $E_\xi$↑ | $F_\beta^w$↑ | $F_\beta$↑ | $\mathcal{M}$↓ |
| **Our lightweight design** (only using RGB) | 35 FPS | 17.87 G | 47.85 MB | 0.879 | 0.657 | 0.745 | 0.065 |
| with additional depth (using RGB and depth) | 21 FPS | 45.16 G | 96.53 MB | 0.885 | 0.670 | 0.762 | 0.062 |

A.6    FUTURE DIRECTION

In this paper, we make the earliest effort to explore deep unsupervised learning in RGB-D SOD, which is demonstrated to achieve appealing performance without involving any human annotations during training. To our best knowledge, this is the first such attempt in RGB-D SOD. Meanwhile, we also investigate several potential directions for future research as follows.

**(1) Exploring on the utilization of depth.** The lack of explicit pixel-level supervision brings new challenge to the RGB-D SOD task, that is, inconsistency and large variations in raw depth maps. In this paper, we address it from two aspects. For depth variations between salient objects and background (*i.e.,* possible inter-class variance), a depth-disentangled network is designed to learn the discriminative saliency cues and non-salient background from raw depth map. For variance within salient objects (*i.e.,* possible intra-object variance), we use thresholding and normalization operations to remove illegal numbers, and apply CRF to smooth the updated pseudo-labels in DLU. One promising future direction is that we can include some post-processing operations to make highlighted region on the "depth-revealed saliency map" uniform before using it to update pseudo-labels. Here we provide an extended experiment in Table 10, where we smooth the inconsistent depth using a new CRF before feeding it to the DLU. This results in higher performance benefiting from more uniform depth. We believe more dedicated algorithm is worth exploring in future work to deal with inconsistent depth.

Table 10: Ablated experiment on the utilization of depth.

| | **NLPR** | | | | **NJUD** | | | |
| --- | --- | --- | --- | --- | --- | --- | --- | --- |
| | $E_\xi$↑ | $F_\beta^w$↑ | $F_\beta$↑ | $\mathcal{M}$↓ | $E_\xi$↑ | $F_\beta^w$↑ | $F_\beta$↑ | $\mathcal{M}$↓ |
| Our DSU framework | 0.879 | 0.657 | 0.745 | 0.065 | 0.797 | 0.597 | 0.719 | 0.135 |
| Our DSU using saliency-guided depth with CRF | 0.885 | 0.661 | 0.751 | 0.064 | 0.799 | 0.603 | 0.724 | 0.133 |

**(2) Exploring on alleviating noise in unsupervised learning.** Noisy problem has always been a universal and inevitable problem for unsupervised learning. In this paper, the proposed ATS strategy is able to alleviate this problem by properly re-weighting to reduce the influence of the ambiguous pseudo-label during training. One promising future direction is that we can design new algorithms to model the uncertainty regions and train the saliency network with partial BCE loss to alleviate the noisy issue of pseudo-labels.

**(3) Exploring on generating high-quality pseudo-labels.** In our paper, a depth-disentangled saliency update (DSU) framework is designed to automatically produce pseudo-labels with iterative follow-up refinements, which is able to provide more trustworthy supervision signals for training the saliency network. For future work, we can explore other algorithms to generate high-quality pseudo-labels, for example, 1) imposing contrastive loss (e.g., using superpixel candidates with ROI pooling to construct contrastive pairs) to enlarge the feature distance between foreground and background; 2) involving partially labeled scribbles to help generate more accurate pseudo-labels.

**(4) Exploring on more fine-grained disentangled framework.** In this paper, we design a depth-disentangled saliency update framework to decompose depth information into saliency-guided and non-saliency-guided depths to promote saliency, motivated by the nature of the class-agnostic binary salient object segmentation

task. It is worthy exploring in the future to design more fine-grained disentangled framework. For example, in addition to the saliency-guided depth and non-saliency-guided depth, we can additionally model the 'not sure' regions by learning an uncertainty map (using well-designed new algorithms such as Monte Carlo Dropout (Gal & Ghahramani, 2016)). The learned uncertainty map, which reveals the 'not sure' regions, can engage in the pseudo-label training process to mitigate the noises of uncertain pseudo-labels, using partial BCE loss.

**(5) Exploring on more applications.** SOD is useful for a variety of downstream applications including image classification (Chen et al., 2021c; Bi et al., 2021; 2022; Ning et al., 2021a;c), light-field data (Zhang et al., 2019; 2020d), medical image processing (Zhao et al., 2021b; Ji et al., 2021b; 2020a; Yao et al., 2021b;a; Ning et al., 2021b; 2020), and video analysis (Zhang et al., 2021; Zhao et al., 2021a), *etc.* This is benefiting from its generic mechanism to highlight class-agnostic objects in a scene. For unsupervised RGB-D SOD, it can be regarded as a pre-task or prior to effectively improve the performance of target task (*e.g.*, weakly supervised semantic segmentation, action recognition), and avoid more laborious efforts.

As discussed above, we summarize five potential research directions for deep unsupervised RGB-D salient object detection. Hopefully this could encourage more inspirations and contributions to this community.

