# OpenReview forum: "Promoting Saliency From Depth: Deep Unsupervised RGB-D Saliency Detection"
_ICLR.cc/2022/Conference — ICLR 2022 Poster_

### Official Review · Reviewer_WhBE · 2021-11-01

**Correctness:** 4
**Technical Novelty And Significance:** 3
**Empirical Novelty And Significance:** 3
**Recommendation:** 8
**Confidence:** 4

**Main Review:**

Overall, the paper is well written and easy to understand.
This paper pointed out the two important issues of unsupervised RGB-D saliency detection; accurate pseudo-label generation and noisy label handling method.
The paper proposed an effective framework to solve the issues and the proposed method consists of two main components; Depth-disentangled Saliency Update (DSU) framework and Attentive training strategy (ATS).
DSU framework generates and refines accurate pseudo-label by utilizing depth map disentanglement and iterative label refinement.
The depth map disentanglement and iterative label refinement complementarily improve each other's quality.
ATS reduces the influence of ambiguous pseudo-label and concentrates on the more reliable training examples by adaptive reweighting strategy.
The contributions of the proposed method are novel. The experimental results including qualitative and quantitative results are impressive.
The extensive ablation study and detailed analysis support the proposed method's effectiveness.

[Minor issue]
1. The description of "Backbone" in the legend of figure 4 is not described in sec 3.3 Attentive training strategy.
It seems the "backbone" is identical with the "backbone" of sec 4.3 Analysis of the empirical result.
I recommend the author to add a description or point another description.


**Summary Of The Paper:**

This paper proposed an effective framework for the unsupervised RGB-D saliency detection task.
The proposed framework consists of two key components. One is the depth-disentangled saliency update framework that is iteratively refined & update the pseudo-labels. The other one is an attentive training strategy that alleviates the influence of noisy pseudo-labels.
The two components mainly tackle the important issues of deep unsupervised RGB-D saliency detection; accurate psuedo-label generation and noisy pseudo-label handling.
Based on the proposed components, the proposed framework achieves state-of-the-art results.


**Summary Of The Review:**

This paper clearly states the problems of unsupervised RGB-D saliency detection and proposed appropriate solutions.
The contribution and novelty are clear. Extensive ablation studiesy reveal the effectiveness of its components.
Based on proposed solutions, the proposed network achieves state-of-the-art results.

---

> ### Author Response · Authors · 2021-11-15
> **Response to Reviewer WhBE**
>
> **We sincerely thank the reviewer for thoroughly reviewing our paper and providing these encouraging comments.
> We have carefully improved the paper according to your suggestions.**
>
> --------------
>
> ***Question1**. The description of "Backbone" in the legend of figure 4 is not described in sec 3.3 Attentive training strategy.
> It seems the "backbone" is identical with the "backbone" of sec 4.3 Analysis of the empirical result.
> I recommend the author to add a description or point another description.*
>
>
> **Answer1**: Thanks very much for pointing out this to us.
> The ‘Backbone’ in Fig. 4 and Sec.4.3 means the saliency network trained with initial pseudo-labels.
> We have added the corresponding description in the caption of Fig.4 in the revised manuscript.
>
> --------------
>
> **Thanks again for your encouragement and suggestions towards improving our paper.
> Please let us know if you have other questions, and we are happy to address them.**

---

### Official Review · Reviewer_kVSL · 2021-11-02

**Correctness:** 2
**Technical Novelty And Significance:** 2
**Empirical Novelty And Significance:** Not applicable
**Recommendation:** 6
**Confidence:** 3

**Main Review:**

Strengths:

1. The idea of decomposing depth cues into saliency-guided depth and non-saliency-guided is novel and makes a good sense, leading to the proposed network architecture and the design of the training loss.
2. The evaluation is extensive and well designed ablation studies are provided.
3. The paper is well organised.

Weaknesses:

1. The core issue is whether the work is an 'unsupervised' method or not given that this is an important claim made in the paper. In my opinion, this work is essentially a weakly supervised method for RGB SOD where the depth information is used as additional knowledge to improve the intial labels generated by the handcrafted method. I also notice that in Table 1, the proposed method is compared with USD and DeepUSPS, which are both unsupervised RGB SOD methods without any involvement of depth information. To this end, the comparison is barely fair as it is well-known that depth information can significantly improve SOD for the methods based only on RGB information.

2. It is stated at the end of Section 3.2 that in the inference stage, "only RGB images are used for predicting saliency". I wonder if the learned mechanism which uses the depth information to refine the RGB-based pesudo label is generalised well on the images not from the same dataset. It might be worthy evaluating the method by training the DSU network on one dataset and then testing it on another one.

3. Despite that the DSU network, particularly the depth-disentangling scheme, offers an effective way to exploit depth information for the improvement of RGB SOD, there is a lack of insightful analysis for the thorough understanding of the scheme. For instance, if we extend the "saliency-guided and non-saliendy-guided" binary taxonomy to a ternary one, would the DSU network performs better? Note that this won't slow down the detection in the inference.

**Summary Of The Paper:**

The paper presents an unsupervised method for RGBD SOD. It internally generates pseudo labels for training by extracting and refining/updating handcrafted features under the framework of ‘depth-disentangling’. Since the updating module can be removed during the inference, it achieves high efficiency.  The method also demonstrates SOTA performance on several benchmarks.

**Summary Of The Review:**

Based on the strengths and weaknesses listed above, I am slightly negative to the paper.

---

> ### Author Response · Authors · 2021-11-15
> **Response to Reviewer kVSL (Part 1/2)**
>
> **We appreciate the time and effort you spent on reviewing our paper.
> The comments and suggestions are valuable and encouraging.
> Here are our detailed replies to your questions.**
>
> --------------
>
> ***Question1**. The core issue is whether the work is an 'unsupervised' method
> or not given that this is an important claim made in the paper.
> In my opinion, this work is essentially a weakly supervised method for RGB SOD where the depth information is used as additional knowledge to improve the initial labels generated by the handcrafted method.
> I also notice that in Table 1, the proposed method is compared with USD and DeepUSPS,
> which are both unsupervised RGB SOD methods without any involvement of depth information.
> To this end, the comparison is barely fair as it is well-known that depth information
> can significantly improve SOD for the methods based only on RGB information.*
>
>
> **Answer1**: In this paper, **we define the new problem of "deep unsupervised RGB-D saliency detection" as:**
> given an unlabeled set of RGB-D images, deep neural network is trained to predict saliency without any laborious human annotations in the training stage,
> as described in the introduction section.
> Therefore, **"unsupervised" here essentially means not using manual annotations, which can greatly save human efforts**.
> We would like to note that RGB-D salient object detection has long been an important research area in the community to explore how to effectively learn from cross-modal RGB-D data.
> A plethora of methods [1-5] have been developed recently in the fully-supervised setting.
> **To avoid laborious manual annotation efforts, our work makes the earliest effort to explore a new problem of deep unsupervised RGB-D saliency detection.**
> For the weakly-supervised setting you mentioned, existing literatures mainly focus on utilizing various low-cost weak supervisions,
> such as image-level category labels, partially-labeled scribbles, bounding box annotations, or image-level captions, to train the saliency network.
> These methods still need human annotation efforts.
>
> As for the comparison experiments in Table 1, since **there is no deep unsupervised RGB-D SOD method available**,
> we additionally provided the performance of two RGB-based deep unsupervised methods, USD and DeepUSPS,
> with clear demarcation to highlight their RGB input nature,
> as described in Table 1 & section 4.2. **Such comparisons are not meant to show the superiority of our method against those methods but only provide observational evidence for the related works**.
> In addition, **to verify the effectiveness of the proposed method**,
> we have conducted extensive ablation analyses (Tables 2, 3, 4, 5, 6 & Figs. 3, 4, 6) and adaptation experiments on fully-supervised setting (Tables 7, 8 & Fig. 9) in the manuscript and appendix.
>
> [1] Rethinking RGB-D salient object detection: models, data sets, and large-scale benchmarks, IEEE TNNLS, 2021.
> [2] RGB-D salient object detection: a survey, Computational Visual Media Journal, 2021.
> [3] Uncertainty inspired RGB-D saliency detection, IEEE TPAMI, 2021.
> [4] Siamese network for RGB-D salient object detection and beyond, IEEE TPAMI, 2021.
> [5] RGB-D salient object detection with cross-modality modulation and selection, ECCV, 2020.
>
>
> --------------
>
>
> ***Question2**. It is stated at the end of Section 3.2 that in the inference stage,
> "only RGB images are used for predicting saliency".
> I wonder if the learned mechanism which uses the depth information to refine the RGB-based pseudo label is generalized well on the images not from the same dataset.
> It might be worthy evaluating the method by training the DSU network on one dataset and then testing it on another one.*
>
>
> **Answer2**: We agree with you that it is worthy evaluating the method by training the DSU network on one dataset and then testing it on another one.
> Actually, we did so in this paper.
> As described in Section 4.1, we follow the setup of previous RGB-D methods [1-3] to construct the training set,
> which includes 1,485 samples from NJUD and 700 samples from NLPR.
> Then **we test our method not only on the remaining images of NJUD and NLPR, but also on two independent test sets including STERE and DUTLF-Depth**.
> Numerical results in Table 1 have demonstrated that our method can generalize well on the images not from the same dataset.
>
> [1] Rethinking RGB-D salient object detection: models, data sets, and large-scale benchmarks, IEEE TNNLS, 2021.
> [2] Calibrated RGB-D salient object detection, CVPR, 2021.
> [3] RGB-D salient object detection via 3D convolutional neural networks, AAAI, 2021.

---

> > ### Author Response · Authors · 2021-11-15
> > **Response to Reviewer kVSL (Part 2/2)**
> >
> >
> > ***Question3**. Despite that the DSU network, particularly the depth-disentangling scheme,
> > offers an effective way to exploit depth information for the improvement of RGB SOD,
> > there is a lack of insightful analysis for the thorough understanding of the scheme.
> > For instance, if we extend the "saliency-guided and non-saliency-guided" binary taxonomy to a ternary one,
> > would the DSU network performs better?
> > Note that this won't slow down the detection in the inference.*
> >
> >
> > **Answer3**: Thanks for your valuable question.
> > Actually, in Sec. 4.3, **we have conducted extensive ablation studies to investigate the contribution of each of the components in our DSU**.
> > In addition, **to further demonstrate the rationality of our method, we have also investigated** "comparisons with other possible pseudo-label generation variants",
> > "the necessity of depth disentanglement", and "how about using raw depth labels in DSU?".
> > **All of these results and analyses consistently demonstrate the superiority of our DSU design**.
> > We are sorry that we do not fully understand what you mean by saying "a ternary one".
> > In general, salient object detection aims to identify the foreground objects and background in a scene.
> > This is actually a class-agnostic binary segmentation task.
> > Motivated by this, we disentangle the depth map into saliency-guided and non-saliency-guided depths in our DSU strategy.
> > If you have further questions, we are happy to address them.
> >
> >
> > --------------
> >
> > **Thanks again for taking the valuable time and providing the insightful comments.
> > Please let us know if you have other questions, and we are happy to address them.**

---

> > > ### Comment · Reviewer_kVSL · 2021-11-23
> > > **Response to ICLR 2022 Conference Paper251 Authors**
> > >
> > > Thanks for the effort of the feedback.  "a ternary one" means that you can have a more fine-grained taxonomy, e.g. reliably salient, not sure, reliably non-salient, and adopt different schemes to each of them.

---

> > ### Comment · Reviewer_kVSL · 2021-11-23
> > **Response to ICLR 2022 Conference Paper251 Authors**
> >
> > Thanks for the effort of the feedback. But I am still not convinced by the "unsupervised" setting adopted in the paper. Since in the inference stage, the proposed method only used RGB image for predicting salinecy, the paper looks like a RGB SOD method. In this case, if it still attempts to leverage the depth information, then in a proper unsupervised  setting, the training should rely only on the RGB images while the (pseudo) depth information can be derived from the images by some depth estimation method. However, in this work, the (ground truth) depth information is directly provided by the dataset. If the authors insist that this is an RGB-D SOD method, then it should be compared with RGB-D SOD methods. If there is no unsupervised RGB-D SOD method available, it should be compared with supervised RGB-D SOD methods and demonstrate that the performance is not significantly degraded.

---

### Official Review · Reviewer_xcAY · 2021-11-02

**Correctness:** 3
**Technical Novelty And Significance:** 3
**Empirical Novelty And Significance:** 3
**Recommendation:** 8
**Confidence:** 4

**Main Review:**

This paper is presented clearly, and the experiments are comprehensive. Additionally, the idea of jointly deriving the saliency training labels from the depth and RGB data is good. However, I have a few questions listed below:

1.  Why discard the available depth signal during inference? This paper discusses RGBD saliency; however, the proposed method is still under the RGB saliency detection framework. The idea will be more impactful if a gated scheme fuses depth information to promote saliency whenever available. Otherwise, the experimental datasets you chose are RGBD, and you only use RGB & Depth to generate the training labels and discard the valid depth signal during inference. Is saving compute the primary reason?

2. When salient objects contain different levels of depth (saying a bus in a frontal view), will the depth distilled saliency/non-saliency signal remain helpful to the pseudo-labels? My concerns come from the observation that D_{Sal} & D_{NonSal} looks similar to the depth map, so when there is a long object with a wide range of depth, the value among the object area in the D_{Sal} won't be uniform; further, if we element-wise multiply such map with the Sal_{pred} in the DLU, will it just make the result even worse?

3. Does error propagation exist in your proposed iterative optimization scheme? What if the depth disentangled saliency signal becomes unreliable, and the iterative optimization scheme guides the network to learn in the wrong direction?

4. When comparing with the handcrafted UnSOD in Table1, do you retrain those methods on your training data to ensure a fair comparison?

5. Inside ATS, given ten epochs, what epoch use step one, and what epoch use step two? Also, is there any stop mechanism besides the current heuristic one?

6. The idea of utilizing the handcrafted saliency method as unsupervised training data and using auxiliary information to refine such training data for the more trustworthy signal is also presented in the following papers:
(1) "Deep co-saliency detection via stacked autoencoder-enabled fusion and self-trained cnns."
(2)"Unsupervised CNN-based co-saliency detection with graphical optimization."
However, the difference separating with them is not discussed in the paper.

**Summary Of The Paper:**

This paper presents an unsupervised RGB saliency detection model to learn from the pseudo-labels given by the traditional handcrafted approach. To enhance the quality of the pseudo labels, the author proposes to disentangle the depth data to promote the saliency signal and compress undesired noise. Last but not least, an attentive training strategy is integrated to teach the network from iteratively updating supervisory labels. Experimental results demonstrate the effectiveness of the proposed depth-disentangled saliency update framework and can help improve the performance of on-the-shelf approaches.

**Summary Of The Review:**

Overall, I think it is currently a borderline paper since the idea is good, which can benefit researchers in related fields. Meanwhile, this work is presented well, and the experiments are comprehensive. However, I hope to check the answer to my questions before making further judgments.

---

> ### Author Response · Authors · 2021-11-15
> **Response to Reviewer xcAY (Part 1/3)**
>
> **Thanks for your positive comments and valuable suggestions. Your questions will be answered point by point.**
>
> --------------
>
> ***Question1.** Why discard the available depth signal during inference?
> This paper discusses RGBD saliency;
> however, the proposed method is still under the RGB saliency detection framework.
> The idea will be more impactful if a gated scheme fuses depth information to promote saliency whenever available.
> Otherwise, the experimental datasets you chose are RGBD,
> and you only use RGB & Depth to generate the training labels and discard the valid depth signal during inference.
> Is saving compute the primary reason?*
>
>
> **Answer1**: Saving computational cost is one of the reasons for discarding depth signal during inference.
> In addition to that, being free of depth stream at test time can also minimize the memory consumption
> (by discarding depth subnetwork and multi-model fusion module) and enables wider practical applications
> (by simplifying the input requirement at deployment stage).
> We agree with you that our idea will be more impactful if a gated scheme is incorporated to decide whether to use depth information.
> We conducted a comparison experiment as shown in the table below,
> comparing the model performance, inference speed, FLOPs and parameter number,
> when using only RGB stream versus using RGB and depth simultaneously.
> It is observed that introducing depth information can further improve the model performance,
> but at the same time, it will lead to 1.7$\times$ decrease of inference speed,
> and more than 2$\times$ increases of FLOPs and parameters.
> Thanks for your valuable suggestion, we will release the source code, pre-trained model and ReadMe.txt,
> which allows researchers to use our model according to their needs.
> These analyses and the experimental results have been added in Appendix (A.5).
>
>
>
> |  	        |                         Complexity                      |                           Performance (NLPR)            |
> |:--------:	|:---------------------------------------------------------:	|:---------------------------------------------------------:	|
> |   	    | Speed / FLOPs / Parameters 	                                | $E_{\xi}$ / $F_{\beta}^{w}$ / $F_{\beta}$ / $\mathcal{M}$ 	|
> |    **Our lightweight design** (only using RGB stream)   	                |    35 FPS / 17.87 G / 47.85 MB   	|   0.879 / 0.657 / 0.745 / 0.065 	|
> | with valid depth  (using RGB, depth and disentangled-depth streams) 	    |    21 FPS / 45.16 G / 96.53 MB    |   0.885 / 0.670 / 0.762 / 0.062 	|

---

> > ### Author Response · Authors · 2021-11-15
> > **Response to Reviewer xcAY (Part 2/3)**
> >
> >
> > ***Question2**. When salient objects contain different levels of depth (saying a bus in a frontal view),
> > will the depth distilled saliency/non-saliency signal remain helpful to the pseudo-labels?
> > My concerns come from the observation that $D_{Sal}$ & $D_{NonSal}$ looks similar to the depth map,
> > so when there is a long object with a wide range of depth,
> > the value among the object area in the $D_{Sal}$ won't be uniform;
> > further, if we element-wise multiply such map with the $Sal_{pred}$ in the DLU,
> > will it just make the result even worse?*
> >
> >
> > **Answer2**: Thanks for your insightful question.
> > The phenomenon of "different levels of depth" you mentioned is formulated as "inconsistency and large variations in raw depth maps" in this paper.
> > We address it from two aspects. **First**, for depth variations between salient objects and background (i.e., possible inter-class variance),
> > a depth-disentangled network is designed to learn the discriminative saliency cues and non-salient background from raw depth map.
> > **Second**, for variance within salient objects (i.e., possible intra-object variance),
> > our DLU design is able to make better use of these non-uniform data with the components in DLU.
> > Specifically, in our DLU, the addition and subtraction operations are less sensitive to non-uniform depth values compared to element-wise multiplication (as shown in the experiment below);
> > moreover, our thresholding and normalization operations can also remove illegal numbers and avoid value overflow;
> > meanwhile, the CRF can further smooth the updated pseudo-labels.
> > Experimental results in Table 4 also demonstrate that our method can significantly improve the quality of pseudo-labels as the training proceeds.
> > **For your question "if we element-wise multiply such map with the $Sal_{pred}$ in the DLU, will it just make the result even worse?"**,
> > we conducted a comparison experiment in the table below.
> > These results reveal that direct multiplication leads to significantly inferior performance compared to our original DLU design since direct multiplication may negatively suppress positive responses in the saliency map.
> > This further confirms the superiority of our DLU design.
> >
> >
> >
> >
> > |  	        |                         NLPR                     |                           NJUD            |
> > |:--------:	|:---------------------------------------------------------:	|:---------------------------------------------------------:	|
> > |   	    | $E_{\xi}$ / $F_{\beta}^{w}$ / $F_{\beta}$ / $\mathcal{M}$  	                                | $E_{\xi}$ / $F_{\beta}^{w}$ / $F_{\beta}$ / $\mathcal{M}$ 	|
> > |    The DLU using Eqs. (5) and (6) of main text (i.e., **Ours**)   |    0.879 / 0.657 / 0.745 / 0.065   	|   0.797 / 0.597 / 0.719 / 0.135 	|
> > |    The DLU using element-wise multiplication in $D_{Sal}$ 	    |    0.851 / 0.590 / 0.656 / 0.095    |   0.780 / 0.547 / 0.665 / 0.162 	|
> >
> >
> >
> > --------------
> >
> >
> >
> > ***Question3**. Does error propagation exist in your proposed iterative optimization scheme?
> > What if the depth disentangled saliency signal becomes unreliable,
> > and the iterative optimization scheme guides the network to learn in the wrong direction?*
> >
> >
> > **Answer3**: When the learned depth saliency signal is unreliable (which will lead to unreliable pseudo-label),
> > **our ATS is able to reduce the influence of the ambiguous pseudo-label**,
> > by properly re-weighting to highlight more reliable ones for training the saliency network.
> > In addition, **our depth-disentangled label update (DLU) and the depth-disentangled network in the DSU are able to complementarily improve each other's quality** with iterative refinement.
> > In detail, when generating the updated pseudo-labels in DLU, the supervisory signals for training the depth-disentangled network are also updated using the improved saliency results.
> > With our iterative training scheme, in the next training round, the depth-disentangled network will in turn produce more reliable disentangled saliency signal for updating the pseudo-labels in DLU.
> > The quantitative & qualitative results in Table 4 & Fig. 6 consistently demonstrate that the pseudo-label quality is gradually improved during the iterative training scheme;
> > the error reduction curve in Fig. 4 also shows significantly improved testing performance by using the proposed DSU and ATS.

---

> > > ### Author Response · Authors · 2021-11-15
> > > **Response to Reviewer xcAY (Part 3/3)**
> > >
> > >
> > > ***Question4**. When comparing with the handcrafted UnSOD in Table1,
> > > do you retrain those methods on your training data to ensure a fair comparison?*
> > >
> > >
> > > **Answer4**: In this paper, we follow the previous RGB-D SOD studies [1-4] to make a fair comparison;
> > > all results are either directly provided by the authors, or generated by re-running their released implementations with default setups.
> > >
> > > [1] Rethinking RGB-D salient object detection: models, data sets, and large-scale benchmarks, IEEE TNNLS, 2021.
> > > [2] RGB-D salient object detection: a survey, Computational Visual Media Journal, 2021.
> > > [3] Uncertainty inspired RGB-D saliency detection, IEEE TPAMI, 2021.
> > > [4] Depth-induced multi-scale recurrent attention network for saliency detection, ICCV, 2019.
> > >
> > >
> > >
> > >
> > > --------------
> > >
> > >
> > >
> > > ***Question5**. Inside ATS, given ten epochs, what epoch use step one, and what epoch use step two?
> > > Also, is there any stop mechanism besides the current heuristic one?*
> > >
> > >
> > > **Answer5**: In Table 3, we analyzed the effect of different alternation intervals $\tau$ in ATS,
> > > where $\tau = 3$ is shown to work best empirically.
> > > For example, given ten epochs, 1 to 3 epochs engaged in step one, 4 to 6 epochs engaged in step two,
> > > amounting to $2\tau = 6$ epochs in a training round.
> > > Then, the proposed DLU kicks in at the end of each training round to update \& refine pseudo-labels.
> > > As the training proceeds, the model moves on to the next training round (i.e., 7 to 9 epochs for step one, ..., etc.).
> > > Other alternative stopping mechanism can be exploited, such as gradient-based scheme [1].
> > >
> > > [1] Early stopping without a validation set, arXiv, 2017.
> > >
> > >
> > >
> > > --------------
> > >
> > >
> > >
> > > ***Question6**. The idea of utilizing the handcrafted saliency method as unsupervised training data and
> > > using auxiliary information to refine such training data for the more trustworthy signal is also presented in the following papers:
> > > (1) "Deep co-saliency detection via stacked autoencoder-enabled fusion and self-trained cnns."
> > > (2) "Unsupervised CNN-based co-saliency detection with graphical optimization."
> > > However, the difference separating with them is not discussed in the paper.*
> > >
> > >
> > > **Answer6**: Thanks for your valuable suggestion.
> > > [1] presents a fusion-learning based approach that jointly explores the image-level confidence and the region-level confidence from co-salient object likelihood,
> > > and employs a self-taught fashion to alleviate over-smoothed saliency maps.
> > > [2] designs two novel unsupervised losses, the single-image saliency loss and the co-occurrence loss, for generating co-saliency maps of high quality.
> > > They both achieve superior performance in addressing co-saliency detection in a set of images jointly covering objects of a specific class.
> > > Different from these approaches, we instead adopt the disentangled depth information to promote pseudo-labels for unsupervised RGB-D SOD task.
> > > We also carefully review related literatures [1-4] and have included them in the revised manuscript.
> > >
> > > [1] Deep co-saliency detection via stacked autoencoder-enabled fusion and self-trained cnns, IEEE TMM, 2019.
> > > [2] Unsupervised CNN-based co-saliency detection with graphical optimization, ECCV, 2018.
> > > [3] Image co-saliency detection and co-segmentation via progressive joint optimization, IEEE TIP, 2018.
> > > [4] Co-attention CNNs for unsupervised object co-segmentation, IJCAI, 2018.
> > >
> > >
> > > --------------
> > >
> > > **Thanks again for your appreciation on our work and providing these valuable suggestions.
> > > Please let us know if you have other questions, and we are happy to address them.**

---

### Author Response · Authors · 2021-11-22
**Thanks for your efforts!**

Dear Reviewers, Area Chairs and Program Chairs,

We sincerely thank the efforts you have made on this paper. The comments from all the reviewers are encouraging and helpful. We have updated the paper accordingly (marked red). Thank you very much.

Best regards,
Authors

---

### Decision · Program_Chairs · 2022-01-20

**Decision:**

Accept (Poster)

**Comment:**

The paper received two accepts and 1 marginally above acceptance recommendations. The authors provided satisfactory answers, mostly on clarifying the unsupervised learning methodology, in conjunction with the MAA recommendation. I recommend the paper be accepted as poster.